# Pulmonary Vascular Underperfusion Score in Premature Infants with Bronchopulmonary Dysplasia and Pulmonary Hypertension

**DOI:** 10.3390/medicina55070359

**Published:** 2019-07-09

**Authors:** Bibhuti B. Das, Michelle-Marie Jadotte, Kak-Chen Chan

**Affiliations:** Joe DiMaggio Children’s Hospital Heart Institute, Memorial Healthcare, Hollywood, FL 33021, USA

**Keywords:** premature infant, bronchopulmonary dysplasia, pulmonary hypertension, digital subtraction pulmonary angiography

## Abstract

Pulmonary hypertension (PH) is a complication of bronchopulmonary dysplasia (BPD). The underlying pathophysiology of BPD-associated PH is complex and poorly understood. Echocardiogram may underestimate the severity of pulmonary hypertensive vascular disease in severe BPD. Digital subtraction pulmonary angiography (DSPA) is a potentially useful imaging modality for evaluating changes in the pulmonary vasculature of BPD-associated PH. In this study, we objectively quantified the pulmonary hypertensive vascular changes demonstrated by DSPA using a novel pulmonary vascular underperfusion score (PVUS) and correlated the scoring system with echocardiography parameters and cardiac hemodynamics by right heart catheterization.

## 1. Introduction

Premature births cause developmental arrest of lungs and impair pulmonary vascular angiogenesis, which decrease the cross-sectional area of pulmonary blood flow and in turn results in bronchopulmonary dysplasia (BPD)-associated pulmonary hypertension (PH). The American Heart Association published guidelines for the management of PH in infants and children [1]. However, specific issues like role of cardiac catheterizations and echocardiography in preterm infants are not addressed in detail. The predictive capacity of echocardiography in the diagnosis of early pulmonary vascular disease in preterm infants is only 33% [2]. Early pulmonary vascular disease is associated with development of BPD and with late PH in preterm infants. We aimed to characterize the morphological changes in pulmonary vasculature by digital subtraction pulmonary angiography (DSPA) in 10 premature infants (≤28 weeks) who have severe BPD and compare the pulmonary vascular underperfusion score (PVUS) with cardiac hemodynamics, and echocardiographic parameters of PH.

## 2. Methods

This was a retrospective, single-center study and included 10 consecutive patients with severe BPD, who were referred to our PH service between January 2016 and December 2018 for further evaluation and management. For the purpose of the study, BPD was defined as high oxygen (≥30% fraction of inspired oxygen (FiO2)) and respiratory support requirements at 36 weeks of corrected gestational age per published guidelines [3,4]. Institutional review board approval (IRB Project No. MHS.2018.116) was obtained on 9 January 2019 to retrospectively review the medical records to obtain demographic profile including gestational age, birth weight, perinatal and post-natal clinical data including duration of oxygen use, duration of ventilator support, echocardiographic and radiographic images. All patients underwent a right heart catheterization (RHC) and informed consent were obtained from parents prior to RHC as per our institutional protocol. Our institutional protocol for evaluation of PH in children routinely included DSPA as a part of RHC instead of standard pulmonary angiography.

The potential advantages of the DSPA technique as previously described include a lower dose of radiographic contrast compared to conventional pulmonary angiography [5]. A Berman catheter was introduced into both branch pulmonary arteries (PAs) and with simultaneous breath-holding, fluoroscopy with dynamic masking was initiated. The first image (the mask) obtained was without contrast, and successive images were obtained after injecting 1 mL/kg of contrast (1:2 diluted). Subsequent images were acquired at 15 frames/s. We compared the total radiation exposure of these 10 patients to 10 heart transplant patients of similar body surface area who also had conventional pulmonary angiography during routine RHC.

The DSPA images for each patient were analyzed for pulmonary vascular underperfusion score (PVUS) by manually counting the areas of perfusion defects: score (0) = pulmonary capillaries seen up to the periphery in uniform manner (Normal); (1) = pulmonary capillary perfusion defect <3 mm from the outer margin of lung parenchyma or patchy defects along the lung periphery, (2) = 3 to <5 mm short from the lung outer margin, (3) = 5–7 mm short from the lung outer margin (winter-tree appearance); and score of 1 for each perfusion defect along the distribution of major inter-lobar branch pulmonary arteries. All scores from both lungs in each subject were then added and a total score was assigned as PVUS for individual patient. Each patient’s PVUS score estimated by 2 observers separately and when there was discrepancy, the observation by senior author (KCC) was the final value used for this study. At the time of RHC, all patients had acute vasoreactivity testing (AVT) with inhaled nitric oxide (iNO) 20 ppm and 100% oxygen. A patient was diagnosed as responder to AVT if the subject experienced a 20% decrease in mean PA pressure and unchanged or improved cardiac index [6].

Echocardiographic images were available in each patient within first 7 days of life and were reviewed. The diagnosis of PH was made by echocardiography based on the following criteria: (1) a tricuspid regurgitation (TR) velocity ≥ 3 m/s in the absence of pulmonary stenosis and/or (2) a flat or left deviated interventricular septal configuration with right ventricular (RV) hypertrophy as previously described [7]. Right ventricular function parameters were also evaluated according to the guidelines of the American Society of Echocardiography [8]. The echocardiogram findings at the time of cardiac catheterization were used to compare the cardiac hemodynamic data for analysis.

## 3. Results

The demographics and clinical data of 10 preterm infants (≤28 weeks gestational age) with severe BPD were summarized in Table 1. All infants had worsening respiratory status and increasing oxygen requirements despite sildenafil therapy having started prior to referral to our service. Seven patients were managed by mechanical ventilation, two were on continuous positive airway pressure (CPAP) and two were on high flow nasal cannula oxygen. All patients’ FiO2 requirement was ≥30%. The median duration of intubation was 45 days. Two of the patients required tracheostomy and chronic ventilation. Initial echocardiogram showed PH based on the tricuspid regurgitation jet ≥3 m/s in six patients only, but there was evidence of ventricular septal flattening in all 10 patients. There were ductus arteriosus (DA) in five and foramen ovale (FO) in six and atrial septal defect (ASD) in two patients. Two of the small DAs closed spontaneously, three patients received ibuprofen therapy and two patients required DA to be closed by interventional procedure.

The hemodynamic parameters were summarized in Table 2. All patients had elevated (>2 WU·m^2^) pulmonary vascular resistance (PVR) but only six patients were reactive to AVT with inhaled nitric oxide (iNO) and 100% oxygen using Barst criteria [6]. There was no significant difference in their mean right atrial (RA) pressure, mean PA pressure, mean indexed PVR (PVRi), PVRi to indexed systemic vascular resistance (SVRi) ratio, and cardiac index between AVT responders and non-responders. We evaluated the DSPA images and analyzed the PVUS for each patient by counting the areas of perfusion defects (Figure 1, Figure 2 and Figure 3). We utilized DSPA for a normal subject without PH to demonstrate the major inter-lobar arteries and to show that there were no perfusion defects (Figure 4). No complications related to RHC and DSPA were observed in our cohort. There was a significantly higher total PVUS (≥4) among four AVT non-responders compared to six AVT responders (≤4) (*p* = 0.0048). Figure 5 describes the mean, the maximum and minimum values and quartiles of PVUS between AVT responders (*N* = 6) and non-responders (*N* = 4) (*p* = 0.005). The intraclass correlation coefficient (ICC) for interobserver variability in estimating PVUS was 0.87, mean difference was 0.67 with 95% confidence interval (CI), −1.3 to +1.3 (Figure 6).

Total radiation exposure in 10 BPD cases was 93.6 ± 30 mGy compared to 82.6 ± 31 mGy (*p* = 0.38) for pediatric heart transplant patients with similar BSA undergoing right heart catheterization and conventional PA angiography as a part of their routine cardiac catheterization (control group). Total fluoroscopy time in patients was 12.5 ± 4.4 min compared to 11.5 ± 2.5 min in the control group (*p* = 0.52). However, the radiation dose and fluoroscopy time were for the total RHC procedure and not for pulmonary angiography alone in both groups.

The summary of echocardiographic parameters of PH and response to AVT were described in Table 2. There were no significant differences in echocardiogram parameters between two groups, and AVT responders and non-responders were not differentiated. Based on PVUS and cardiac hemodynamic findings, bosentan therapy was added to sildenafil for four of our patients who were AVT non-responders. Subsequently, all four patients had improvement in their oxygen requirement and echocardiographic parameters of PH.

## 4. Discussion

In our study, we have shown that PVUS ≥4 by DSPA has improved the accuracy of recognition of AVT responders vs. non-responders. Based on higher PVUS scores in four patients, bosentan was added in addition to sildenafil for PH therapy. Subsequently, the combined bosentan and sildenafil therapy resulted in decreased oxygen requirement and improved clinical outcomes. Sildenafil which is often used as a first line therapy may not be effective if there is significant ventilation/perfusion mismatch secondary to pulmonary perfusion defects [9]. Combination therapy (sildenafil and bosentan) may be effective in patients not responding to single vasodilator [10].

Impaired angiogenesis during alveolar development leading to decreased vascular surface area, pulmonary arterial remodeling, and increased PVR are the proposed pathophysiological mechanisms involved in BPD-associated PH [11]. Consequently, a better understanding on how alveoli and the underlying capillary network develop and how these mechanisms are disrupted in BPD is critical to develop efficient and effective therapies to prevent lung injury or regenerate established lung injury in premature infants. Recent meta-analyses found that the presence of PH in BPD patients strongly associated with a high mortality rate (40%) during first two years of life [12]. No trials have yet been conducted to formally assess the effect of PH-specific therapies on children with BPD. However, effective use of current available therapies is essential to maximize the benefit. Recent guidelines have been published to help standardize provider approach, diagnosis, monitoring, and management of PH in children with BPD [13,14]. We speculate that, the lack of improvement in PH in some infants with BPD may be due to advanced pulmonary hypertensive vascular disease due to delay either in diagnosis or initiation of combination therapy. Referral to PH experts for early RHC and in selected patients DSPA may aid in management and prognostication. Multicenter and prospective studies are required to evaluate the role of DSPA as imaging modalities and the role of PVUS in long-term management of PH in this challenging group.

Limitations: The DSPA technique may entail slightly higher fluoroscopy time and total radiation exposure compared to standard pulmonary angiogram. It is not possible to have a direct comparison of DSPA technique versus conventional PA angiography in normal control population. In this study we compared total radiation exposure for RHC including conventional PA angiogram versus RHC including DSPA, the total radiation dose and fluoroscopy time were not statistically significant. The other disadvantage of the DSPA technique was the need for controlled breath holding during the acquisition of images.

## 5. Conclusions

Digital subtraction angiography is useful to demonstrate the morphological changes of pulmonary vasculature in what is traditionally thought of as an airspace disease and to open new therapeutic avenues to treat PH in BPD patients. The novel use of PVUS in a very small cohort of selective premature infants with severe BPD helped us to diagnose advanced pulmonary vascular changes and initiate combination therapy (bosentan and sildenafil) with subsequent clinical improvement, but our findings may not be generalized to all BPD patients. We speculate that some BPD infants who do not respond to sildenafil may have developed advanced changes in pulmonary vasculature. Large multicenter studies are required to confirm our findings that PVUS can be used to risk stratify patients with BPD-associated PH.

## Figures and Tables

**Figure 1 medicina-55-00359-f001:**
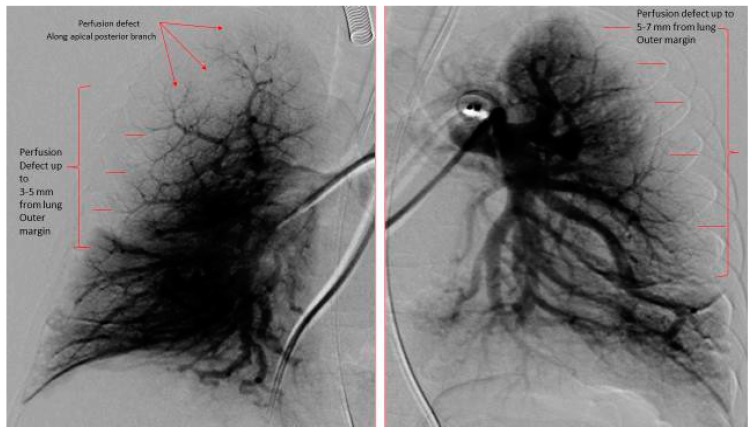
Digital subtraction pulmonary angiography of both right and left lungs. Right Lung: Pulmonary vascular underperfusion score (PVUS) = 4 (perfusion defect from 3–5 mm from the outer lung margin (2) plus perfusion defect of apical posterior branch and anterior branches (2)). Left Lung: PVUS = 2 (perfusion defect from 5–7 mm from the outer lung margin). Total PVUS for this patient = 6.

**Figure 2 medicina-55-00359-f002:**
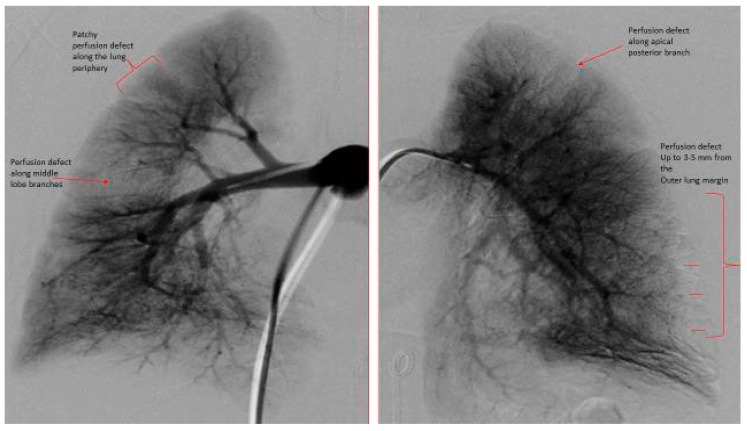
Digital subtraction pulmonary angiography of both right and left lungs. Right Lung: PVUS. = 2 (patchy perfusion defect along the outer lung margin (1) plus perfusion defect of middle lobe branches (1)). Left Lung: PVUS = 2 (perfusion defect from 3–5 mm from the outer lung margin (1) plus perfusion defect along apical posterior branch (1)). Total PVUS for this patient = 4.

**Figure 3 medicina-55-00359-f003:**
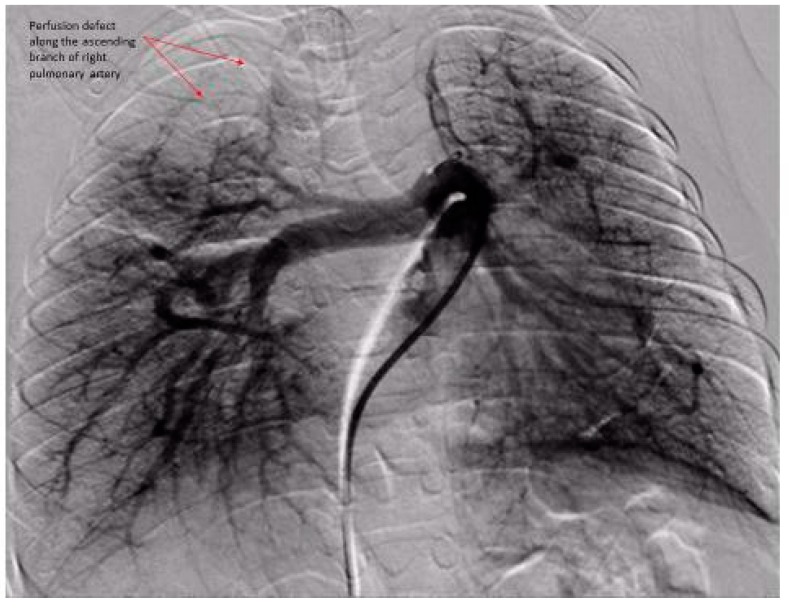
Digital subtraction pulmonary angiography of both right and left lungs. Right lung: PVUS = 2 (perfusion defect of the ascending branch of right pulmonary artery along both apical anterior and posterior branches). Left lung: PVUS = 1 (there are patchy perfusion defect along the outer margin of the left lower lobe up to 3 mm from the outer margin). Note: The significant pulmonary artery dysgenesis with abnormal branching pattern in both lungs. Total PVUS for this patient = 3.

**Figure 4 medicina-55-00359-f004:**
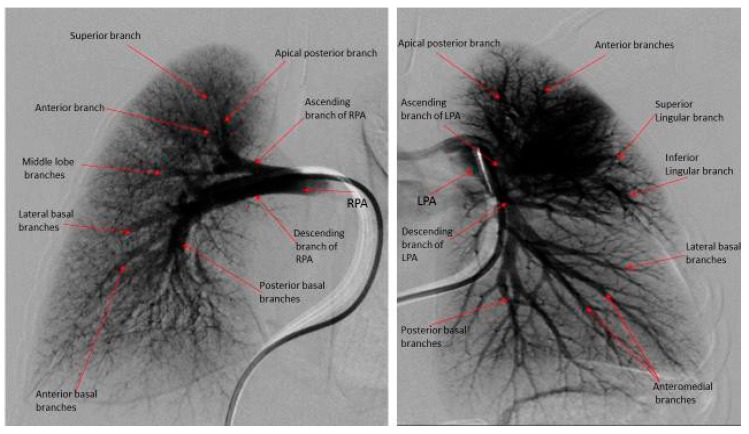
Digital subtraction pulmonary angiography of an eight-month old boy with normal pulmonary artery pressure and pulmonary vascular resistance. Note the digital subtraction pulmonary angiography of both right and left lungs have normal branching pattern, uniform capillary arborization with a clean peripheral lung margin. Total PVUS = 0.

**Figure 5 medicina-55-00359-f005:**
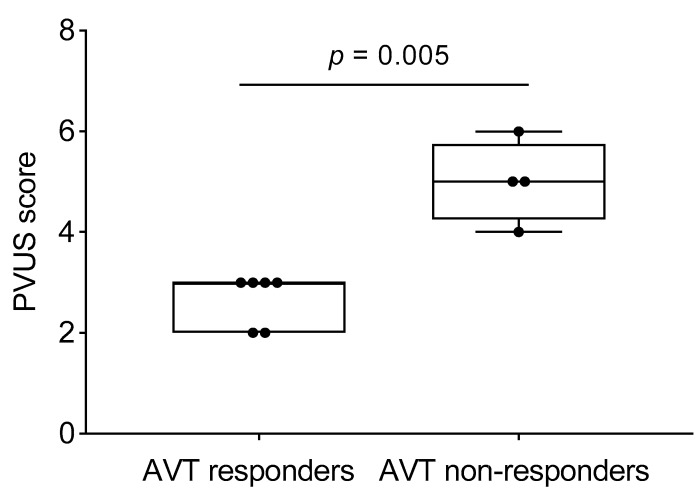
PVUS between acute vasoreactivity testing (AVT)-responders versus non-responders (Mean, minimum, maximum and quartiles).

**Figure 6 medicina-55-00359-f006:**
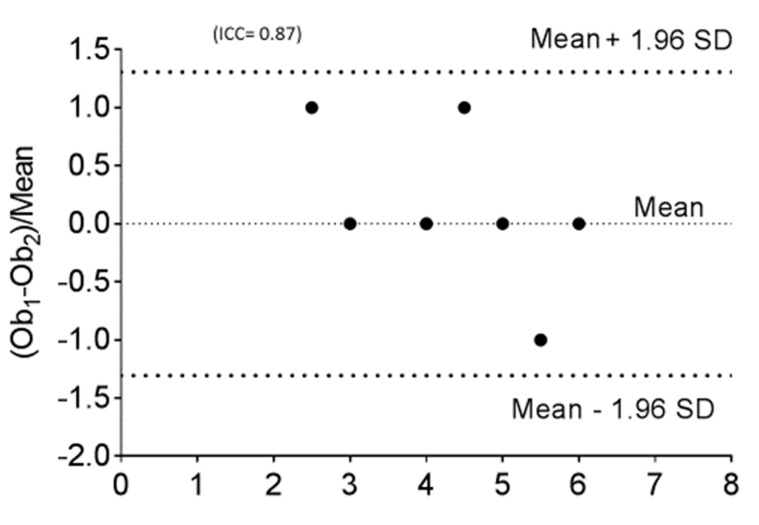
Bland-Altman plot for interobserver variability for estimating PVUS in all 10 cases (three cases have exactly the same value, so in the plot only seven points are represented).

**Table 1 medicina-55-00359-t001:** Summary of maternal and infant characteristics.

Cases (*N* = 10)	
Gestational age (weeks ± SD)	25.1 ± 2.02
Birth weight (grams ± SD)	706 ± 240
SGA (*n*) (%)	3 (30)
Gender: Male, *n* (%)	4 (40)
C-section delivery, *n* (%)	7 (70)
Antenatal steroid, *n* (%)	9 (90)
Preeclampsia (Yes), *n* (%)	4 (40)
HELLP (Yes), *n* (%)	1 (10)
Pregnancy-induced hypertension (Yes), *n* (%)	1 (10)
Chronic hypertension (Yes), *n* (%)	3 (30)
Oligohydramnios (Yes), *n* (%)	1 (10)
Polyhydramnios (Yes), *n* (%)	1 (10)
Clinical chorioamnionitis (Yes), *n* (%)	0 (0)
GBS (+), *n* (%)	1 (10)
APGAR at 5 min, (minutes ± SD)	5.6 ± 1.2
Received surfactant after birth (Yes), *n* (%)	5 (50)
PDA (Yes), *n* (%)	5 (50)
PFO/ASD (Yes), *n* (%)	6 (60)
Intubation after birth), *n* (%)	7 (70)
Duration of intubation (days ± SD)	45 ± 38
Tracheostomy, *n* (%)	2 (20)
Necrotizing enterocolitis (Bell’s stage 2 +), *n* (%)	4 (40)
Severe intraventricular hemorrhage, *n* (%)	2 (20)
Retinopathy of prematurity, *n* (%)	4 (40)

SGA, small for gestational age, HELLP, hemolysis, elevated liver 4 enzymes and low platelet count, GBS, group B streptococci, APGAR, appearance, pulse, grimace, activity, respiration, PDA, patent ductus arteriosus, PFO, patent foramen ovale, ASD, atrial septal defect.

**Table 2 medicina-55-00359-t002:** Echocardiogram and hemodynamic characteristics according to response to acute vasodilator testing (AVT).

Variables	AVT Responder (*N* = 6)	AVT Non-Responder (*N* = 4)	P
Right heart catheterization:			
Age at cardiac catheterization (months ± SD)	9.6 ± 2.8	9.7 ± 2.8	0.96
Weight at cardiac catheterization (kg ± SD)	3.9 ± 0.64	4.4 ± 0.6	0.28
Mean RAP mmHg (mean ± SD)	11 ± 4.7	9 ± 1.3	0.31
Mean PAP mmHg (mean ± SD)	40 ± 7	45 ± 7	0.32
Mean PVRi (WU·m^2^) (mean ± SD)	3.3 ± 2.5	3.5 ± 2.3	0.13
Mean SVRi (WU·m^2^) (mean ± SD)	9.8 ± 9.5	10.2 ± 6.8	0.53
PVRi/SVRi (mean ± SD)	0.32 ± 0.1	0.34 ± 0.04	0.71
Mean Cardiac Index (L/min/m^2^) (mean ± SD)	3.6 ± 1.3	3.9 ± 1.6	0.45
Digital subtraction pulmonary angiogram:			
Pulmonary Vascular Underperfusion Score (PVUS) ^a^ (mean ± SD)	2.66 ± 0.47	5 ± 0.81	0.0048
Echocardiographic parameters (not done with simultaneous cardiac catheterization)			
^b^ RVSP based on TR jet mmHg (mmHg ± SD)	40 ± 4	53 ± 10.6	0.06
^c^ RV Fractional shortening area change % (mean ± SD)	0.39 ± 0.09	0.31 ± 0.09	0.07
^d^ TAPSE (cm ± SD)	12.4 ± 1.6	10.5 ± 1.9	0.15
^e^ AT/RVET (mean ± SD)	0.33 ± 0.02	0.29 ± 0.13	0.13
^f^ RV/LV ratio (mean ± SD)	0.94 ± 0.05	0.96 ± 0.04	0.89

RAP, right atrial pressure, PAP, pulmonary artery pressure, PVRi, pulmonary vascular resistance index, SVRi, systemic vascular resistance index, RVSP, right ventricular systolic pressure, TR, tricuspid regurgitation, RV, right ventricle, LV, left ventricle, SD, standard deviation. ^a^ PVUS is described in methods section. ^b^ RVSP based on TR jet was inadequate to estimate based on TR jet in six cases. RV systolic pressure = TR gradient + mean right atrial pressure. ^c^ RV Fractional shortening area: RV end diastolic area-RV end systolic area/RV end diastolic area × 100. ^d^ TAPSE: Tricuspid annular plane systolic excursion measured in mm from end-diastole to end-systole using M-mode of tricuspid annulus. ^e^ AT/RVET: The ratio of time to peak acceleration time (AT) and right ventricular ejection time (RVET). ^f^ RV/LV Ratio: Ratio of RV diameter and LV diameter at end-systole (parasternal short axis view of the right and left ventricle at the level of papillary muscle.

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
