# Peer review of "Pulmonary Vascular Underperfusion Score in Premature Infants with Bronchopulmonary Dysplasia and Pulmonary Hypertension"

_medicina, 2019, doi:10.3390/medicina55070359_

Round 1
Reviewer 1 Report
1) A well designed study of important significance.
2) While DPSA may be a great resource in PH evaluation, irradiation remains a major concern
3) The authors concluded that DPSA may be useful in differentiating AVT responders from non- responders. Neonatologists/cardiologists familiar with PH management would not wait for DPSA results to label a patient a non-responder or add a second drug. The real question is whether the PVUS scores correlated with the BPD/PH severity/classififcation, which in my opinion, is a working progress.
Author Response
Thank you. We appreciate the comments from the esteemed reviewer.
2. Agree irradiation is concern for DPSA versus conventional pulmonary angiography. Right heart catheterization is the gold standard to diagnose and manage PH. In BPD patients, because the patients are premature have and small body weight, cardiac catheterization is avoided. But when it is necessary to add further management, DPSA is technique which is routinely done at our center for evaluation of PH in all patients. The details of our technique has been described in our previous paper (Ref: Das B, Jadotte M, Mills J, Chan KC. Digital subtraction angiography in children with pulmonary hypertension due to bronchopulmonary dysplasia. Medical Sciences. February 2019). In the revised manuscript, we calculated the total radiation dose in 10 BPD cases included in the study and compared to 10 control patients who had only RHC with conventional PA angiograms, we found the radiation dose and fluoroscopy time was not significantly different. However, the limitation to do head-to-head control is added in limitation section.
3. This is a great point. We are dealing with selected BPD patients who have severe BPD . It will be biased to compare our data with severity of BPD from this cohort. But the reviewer has made a very good point, if this can be doe in a large number of patients from multi-center study, it will be very useful. Thank you
Reviewer 2 Report
1. For the sake of completeness, the method/procedure for digital subtraction pulmonary angiography should be described in this study.
2. Please briefly describe the evaluation/diagnosis of PH and BPD for these patients. Was the diagnosis of BDP associated PH based on echo or cath?
3. In DSPA, is the outer lung margin clearly defined? Do you fix a number of measurements for the distance between the margin and the perfusion defect?
4. Was PVUS given by one clinician only? Was he/she blinded to AVT results? I am wondering how the intra-operator repeatability and inter-operator reproducibility affect the results.
5. In Table 2, mean PVRi=3.3 and mean CI=3.6. Thus, the transpulmonary pressure gradient is ~ 3.3x3.6=11.88 mmHg. Given the mean PAP of 40 mmHg, mean PCW is ~28 mmHg which looks abnormal for these patients. How do you interpret these cath data?
6. Since there are 10 patients in this study, it would be more appropriate to report median values instead of mean values. Box-whisker plots for PVUS can better represent the data distribution compared to the mean values.
7. I am wondering how PVUS correlates with the severity of PH and responses to AVT.
8. Page 6 line 131, “In our study, we have shown that the use of DSPA in conjunction with RHC improved the accuracy of recognition of AVT responders vs. non-responders and improved management decisions and treatment outcomes.” In this study, AVT was already performed. What is the benefit of DSPA analysis to improve the identification of AVT responders? The results showed a significant difference in mean PVUS value between AVT responders and non-responders only and did not support the claims of improved management decisions and outcomes.
Author Response
Thank you.
1. We described our DPSA technique and added to the revised manuscript.
2. In the methodology, paragraph-2, lines 53-58: we described that diagnosis of PH was done by echocardiography in all patients. When these patients have continued requirement of oxygen and needed further evaluation, they have undergone right heart catheterization. These are 10 selected patients referred to us. It is difficult to generalize our findings to all BPD patients.
3. The outer margin is sometimes difficult to define as the perfusion defects are more notable at the periphery (capillary level).
4. The PVUS scoring was done independently by two observers: BD and KCC. We added inter-observer (BD) vs KCC variability to report the ICC and Bland-Altman plot in the revised manuscript (Figure-5).
5. Reviewer again raised a very important question. The trans pulmonary gradient calculated by reviewer is 28, which is due to use of mean value. Cardiac index ranged from 2.6 to 7.5, PVRi range from 2 to 7.5 at baseline. Similarly there was a range of CI and PVRI after 100% oxygen and iNO. There was no elevation of PCWP or left heart disease in our patients.
6. Agree with reviewer, we present the Box-plot for PVUS score. Figure-4 was added to the revised manuscript.
7. In Table-2, there was no difference in PVRI absolute value between responders and non-responders. We used Barst critetria for AVT response (Ref #6: reduction of mean PAP by 20% from baseline with a stable or improved CI) in stead of Sitbon criteria. The PVUS was more than 4 in all AVT non-responders.
8. In the sentence on question: we found that PVUS score was helpful to differentiate AVT responders from non-responders well where as PVRi could not. PVUS more than or equal to 4,able to differentiate all AVT-responders from non-responders. From our limited experience, we suggest, PVUS was helpful to differentiate responders and non-responders as an adjunct to Barst criteria and help to initiate Bosentan with good clinical outcome. Again, this is for a selected very small cohort (more severe form of BPD), although consecutive cases referred to us. And our findings may not be generalizable to all BPD patients.